# Transcriptional profiling of canine osteosarcoma identifies prognostic gene expression signatures with translational value for humans

Joshua D. Mannheimer [1], Gregory Tawa[2], David Gerhold[2], John Braisted[2], Carly M. Sayers [3], Troy A. McEachron[3], Paul Meltzer [4], Christina Mazcko[1], Jessica A. Beck [1] & Amy K. LeBlanc [1✉]

Canine osteosarcoma is increasingly recognized as an informative model for human osteosarcoma. Here we show in one of the largest clinically annotated canine osteosarcoma transcriptional datasets that two previously reported, as well as de novo gene signatures devised through single sample Gene Set Enrichment Analysis (ssGSEA), have prognostic utility in both human and canine patients. Shared molecular pathway alterations are seen in immune cell signaling and activation including TH1 and TH2 signaling, interferon signaling, and inflammatory responses. Virtual cell sorting to estimate immune cell populations within canine and human tumors showed similar trends, predominantly for macrophages and CD8+ T cells. Immunohistochemical staining verified the increased presence of immune cells in tumors exhibiting immune gene enrichment. Collectively these findings further validate naturally occurring osteosarcoma of the pet dog as a translationally relevant patient model for humans and improve our understanding of the immunologic and genomic landscape of the disease in both species.

[1] Comparative Oncology Program, Center for Cancer Research, National Cancer Institute, National Institutes of Health, Bethesda, MD, USA. [2] Division of Preclinical Innovation, Therapeutic Development Branch, National Center for Advancing Translational Sciences, National Institutes of Health, Rockville, MD, USA. [3] Pediatric Oncology Branch, Center for Cancer Research, National Cancer Institute, National Institutes of Health, Bethesda, MD, USA. [4] Genetics Branch, Center for Cancer Research, National Cancer Institute, National Institutes of Health, Bethesda, MD, USA. ✉email: Amy.leblanc@nih.gov

Osteosarcoma (OS) is a rare primary skeletal malignancy of childhood and adolescence, affecting less than 1000 patients per year in the US[1–4]. Osteosarcoma also develops spontaneously in pet dogs, accounting for more than 85% of all canine skeletal malignancies and totaling at least 10,000 new cases in the US alone every year[5–7]. In both species, the biological behavior of OS is aggressive, exhibiting a high propensity for metastasis[8,9]. As outcomes for both species have stagnated for decades, unmet clinical needs have spurred interest in exploring pet dogs with OS as informative models for human OS[10–14]. Comparative oncology studies in tumor-bearing pet dogs are conducted to answer key research questions regarding drug target discovery, and shared tumor biology, and to explore novel avenues for therapeutic optimization[15]. In particular, the comparative approach has been effective in the context of drug development, spanning small molecules, biologics, immunotherapies, imaging agents, and combination strategies[13,15–19]. However, as precision medicine and immuno-oncology come to the forefront of cancer drug research, a detailed molecular framework for canine osteosarcoma is needed to further determine its translational relevance to humans and to facilitate the discovery and validation of new druggable targets[20]. Canine OS recapitulates many of the hallmark biologic and molecular features of human OS, such as highly rearranged genomes with extensive copy-number aberrations and localized hypermutation[21–23]. Like human OS, canine OS genomes generally do not possess high-frequency activating/inactivating mutations in canonical oncogenes and tumor suppressor genes, but rather somatic copy number alterations at genomic loci encoding these genes. The most frequently altered genes in canine OS include *TP53*, *CDKN2A*, and *RB1*[21,24,25].

Alongside the knowledge that describes the genomic complexity of OS comes the challenge of identifying and utilizing appropriate preclinical models for prioritization of the most promising treatments for both humans and canines. While patient-derived xenografts and parallel clinical trials using murine avatars have been used to identify and functionally validate various molecular therapeutic targets, the lack of an intact immune system reveals the inadequacy of utilizing these preclinical models to test combination immunotherapies[26]. The study of naturally occurring OS in dogs can fill gaps in preclinical disease modeling while also providing insight into the biology of a common canine malignancy, given that OS tumors in both humans and dogs develop and progress spontaneously alongside a co-evolving tumor microenvironment and an intact, educated immune system[27,28]. This becomes particularly germane given that results of recent human clinical trials evaluating antibody-based checkpoint inhibitors, CAR-T cells, and immunostimulants have been disappointing and/or difficult to interpret or generalize[29,30]. Combining targeted molecular therapies and/or chemotherapies with relevant immunomodulatory agents presents a new frontier in treating a disease that currently lacks a clear set of druggable driving events.

As in humans, biologic samples collected from canine patients enrolled in clinical trials have the potential to inform the field by virtue of their standardized diagnostic and therapeutic regimens and deep clinical annotation. One clear advantage of canine cancer patients in this context is the ability to enroll treatment-naïve tumor-bearing dogs into therapeutic clinical trials that evaluate investigational agents, providing access to patient materials that represent the ground truth of the disease and a molecular landscape that is free of treatment-induced perturbations. The work presented here leverages a large cohort of canine OS samples procured from a prospective, randomized clinical trial in which over 300 dogs underwent standardized therapy and clinical monitoring. This dataset is the first of its kind that could determine if transcriptional profiling of canine OS can identify distinct molecularly defined patient subsets and/or prognostic gene signatures among dogs receiving standardized therapy, and evaluate if and how these signatures are translatable to human OS using publicly available data.

Herein we provide strong evidence to establish that transcriptomic and clinical patterns identified in canine OS patients could be applied comparably to human OS. Further analysis revealed shared cellular processes in both species, suggesting a transcriptional program that exhibits a rich interplay between tumor cells and immune cells in the tumor microenvironment. This provides a springboard for future investigations of comparative single-nuclei transcriptomics, epigenomic profiling, and geospatial transcriptional studies. Further, evidence of shared cellular processes and associated biology between canine and human OS primary tumors underpins the value of the dog as a patient model for human OS and provides opportunities to explore these pathways as druggable targets in both species. Further, this dataset allowed the assessment of immune-related gene signatures, verification of immune cell subtypes within primary OS tissues, and determination of whether immune cell quantification within tumors can serve as a surrogate for gene signature cluster assignment. Moving forward, these data will assist in the identification of subsets of dogs that may serve as ideal candidates for specific therapeutic testing. For example, analysis of immune-related gene signatures may identify subsets of dogs more apt to respond to specific immunotherapies; or may indicate that certain immune-targeted therapies could be rationally combined to improve outcomes. With this understanding, we gain a greater appreciation of the comparative biology of canine and human OS and insight into the factors that impact patient outcomes.

## Results

**Canine-derived gene signatures are prognostic in both canine and human OS.** To evaluate the clinical significance of two previously identified gene signatures (GS-1 and GS-2)[21] we applied these signatures to the National Cancer Institute's DOG² dataset and performed K-means clustering (Fig. 1). Independent of clinical outcome, data was partitioned into two distinct clusters defined by expression of GS-1 genes (Fig. 2a). Further analysis revealed that the increased enrichment of GS-1 was associated with significantly improved DFI (Fig. 2b) and OSv (Fig. 2c) when compared to the cluster defined by decreased GS-1 enrichment. As such, these GS-1 clusters were subsequently referred to as the favorable prognosis (FP) and poor prognosis (PP) groups, respectively. Similarly, K-means clustering of GS-2 formed clusters (Fig. 2d) that significantly correlated with DFI (Fig. 2e) and OSv (Fig. 2f).

To evaluate the relevance of the canine GS-1 and GS-2 signatures to human OS we applied the same analysis to the human TARGET dataset. As in dogs, K-means clustering of the GS-1 signature resulted in a favorable prognosis and poor prognosis group as it pertains to progression free survival (PFS) and overall survival (Fig. 3a–c). However, the groups resulting from clustering of the GS-2 signature held significance for overall survival only (Fig. 3d–f). Next, we investigated how the stage of disease influenced the overall cluster composition within the TARGET dataset, which contains data derived from primary tumors of human patients with ($n = 20$) and without ($n = 59$) evidence of macroscopic metastases at diagnosis. With the application of both GS-1 and GS-2 signatures to non-metastatic patients, no significant differences in PFS or overall survival were seen (Supplementary Fig. 1). However, for patients presenting with metastasis, the GS-1 signature places 5 patients in the FP

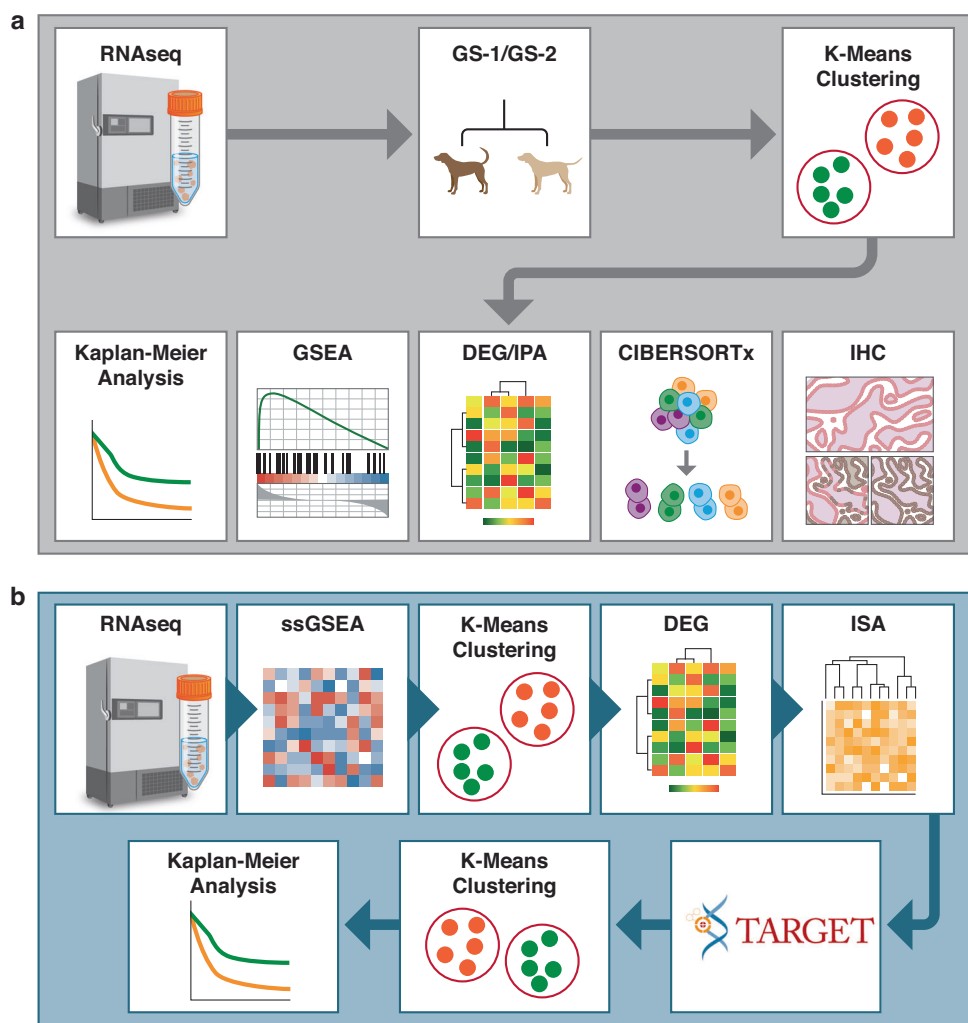

**Fig. 1 Computational approach to mRNAseq datasets. a** General workflow for application of GS-1, GS-2 gene signatures, and downstream analyses. mRNA is filtered down to a gene expression signature followed by K-Means clustering independent of any associated outcome data. Clusters are subjected to several modes of secondary analysis to discern differences in clinical outcome, differentially expressed genes and pathways, and deconvolution of immune cell types with immunohistochemical validation in tissue. **b** General workflow for Iterative Search Algorithm (ISA) signatures devised from canine DOG[2] cohort mRNAseq data. K-means clustering is performed on ssGSEA results that are used as input to a secondary DEG analysis. ISA bi-clustering is then performed on the DEGs, creating new gene signatures. These new ISA signatures are then applied to the human TARGET dataset to determine if the signatures define clusters with meaningful differences in survival.

group and 15 clustered in the PP group (Supplementary Fig. 2a). As expected, both progression free and overall survival were significantly different (Supplementary Fig. 2b, c). With GS-2 applied to metastatic samples, 4 patients clustered in the FP group and 16 in the PP group (Supplementary Fig. 2d). The progression free survival was not significant ($p = 0.081$), but the overall survival was ($p = 0.044$) (Supplementary Fig. 2e, f). This finding upholds the long-recognized impact of the presence of metastatic disease at diagnosis as the single most predictive factor for both canine and human patients with OS[31], and underscores the need to more clearly define mechanisms that link transcriptional programs and other genomic processes within primary tumors to the biology of metastatic progression[8,14,32].

**Transcriptional clusters are enriched for specific cellular processes.** To provide contextual relevance of the gene signatures, we performed GSEA using the msigDB hallmarks version 7.5 signatures[33]. Using the clusters formed using GS-1, 10 pathways in the DOG[2] (Fig. 4a) and 12 pathways in the TARGET data (Fig. 4b) were significantly enriched in the favorable

prognosis (FP) group. Furthermore, all 10 significantly enriched pathways in DOG[2] FP group were also significant in the TARGET FP group, including pathways related to immune function and inflammation such as interferon alpha/gamma, complement system, IL2- STAT5 signaling, Kras[34], and tumor necrosis factor signaling. If only the non-metastatic TARGET samples were considered, 11 of the 12 total pathways had an FDR of less than 0.05 (Fig. 4c) enriched in the FP group. Likewise, only 7 pathways had a significant FDR in the TARGET metastatic-only cohort (Fig. 4d). The GSEA analysis based on the clusters given by GS-2 had all the same significant pathways as GS-1 but included the reactive oxygen species pathway for all three datasets and apoptosis for DOG[2] (Supplementary Fig. 3). No pathways were significantly enriched in the poor prognosis (PP) group for either signature (Supplementary Figs. 4, 5).

**Differential expression analysis points to overlapping pathways in canine and human OS.** Differential expression analysis followed by pathway analysis again pointed to overlapping

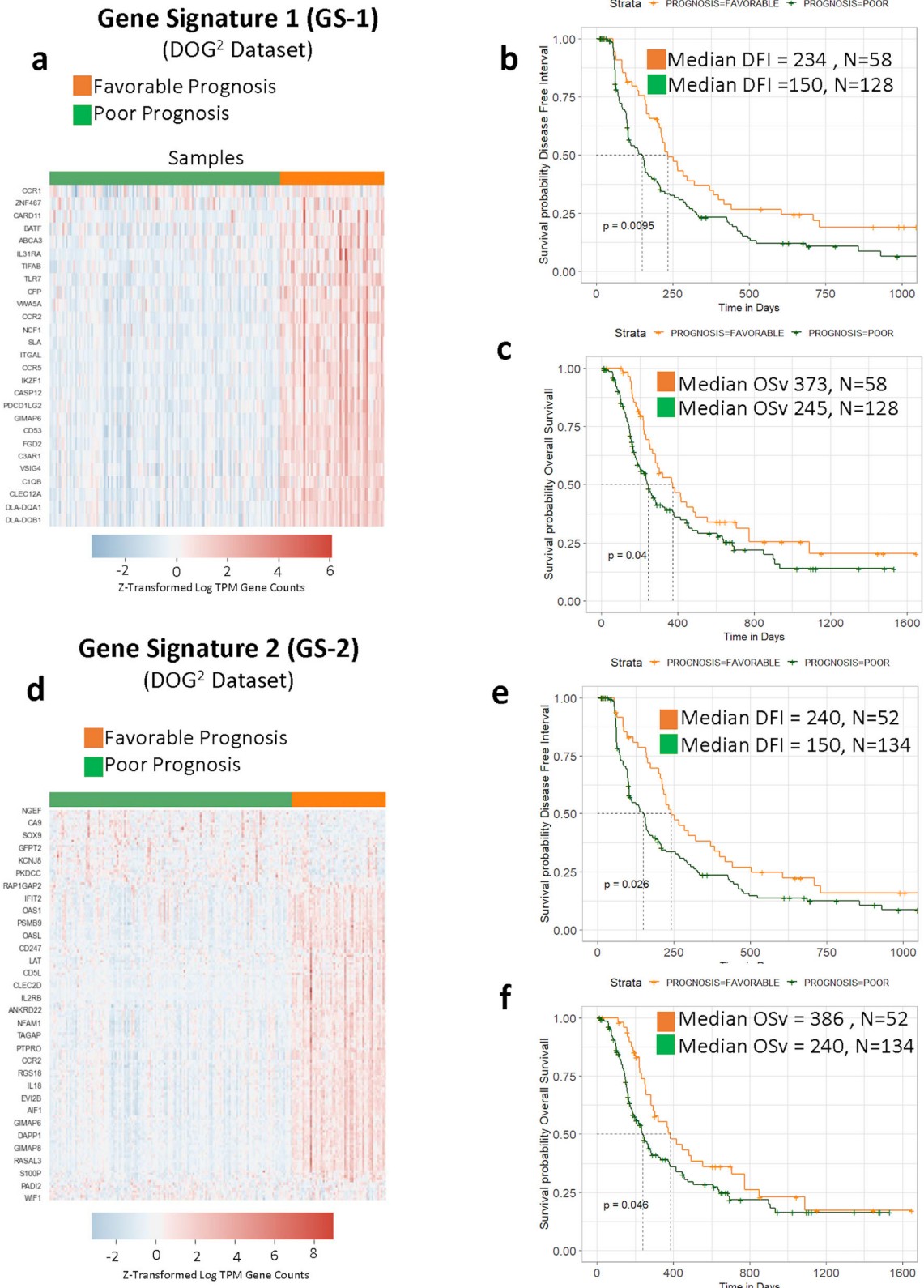

**Fig. 2 Canine osteosarcoma gene signatures are prognostic within the DOG² dataset. a** Expression profile across GS-1 clusters on DOG² data. The two distinct clusters identified by GS-1 have significantly different Kaplan–Meier curves for disease free interval (DFI, **b**) and overall survival (OSv, **c**), both given in days from diagnosis. Expression profiles across GS-2 clusters on DOG² data (**d**). The clusters identified by GS-2 demonstrate significantly different DFI (**e**) and OSv (**f**).

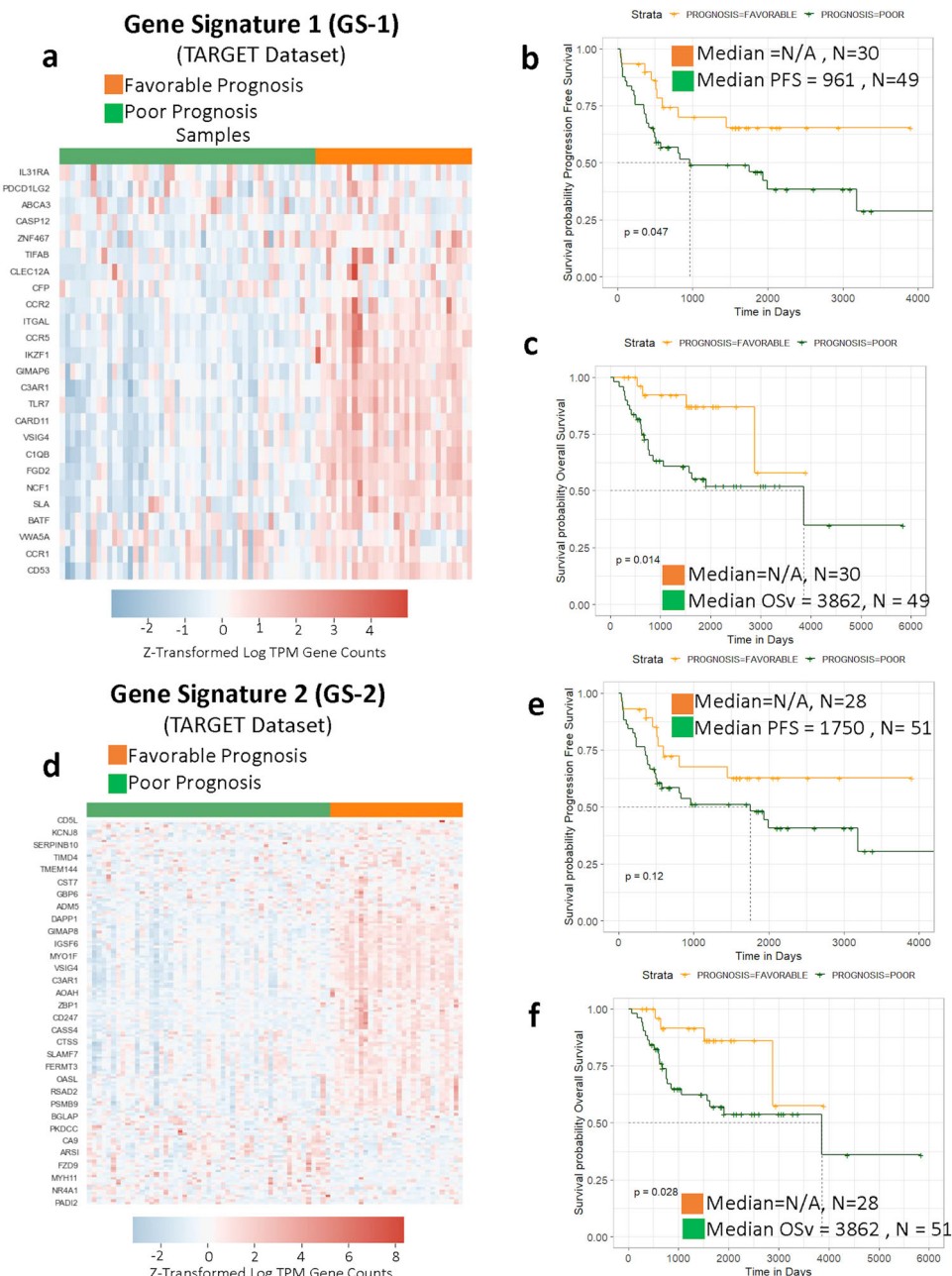

**Fig. 3 Canine osteosarcoma-derived gene signatures are prognostic in human osteosarcoma.** Expression profile of the canine-derived gene signature GS-1 applied to clusters on TARGET data (human osteosarcomas) (**a**). The groups identified by GS-1 have significantly different Kaplan–Meier curves for progression free survival (PFS, **b**) and overall survival (OSv, **c**). **d** The expression profile of the canine-derived gene signature GS-2. In contrast to GS-1, GS-2 clusters do not have a distinct difference in PFS (**e**) but do demonstrate significant differences in OSv (**f**). Median PFS and OSv is given in days from diagnosis.

biological processes and pathways relevant to distinguishing favorable and poor prognosis groups formed by GS-1. For DOG[2] there were 317 DEGs, 220 (69.4%) which were not represented in GS-1 or GS-2. For TARGET, including all 79 samples, there were 281 DEGs, 233 (83%) not in included in GS-1 or GS-2. Likewise, for the 59 non-metastatic samples there were 321 DEGs, 270 (84.1%) not included in GS-1 or GS-2. In the 20 metastatic samples, there were a total of 9 DEGs. Five overlapped with the 79 patient TARGET dataset (IL2RB1, MSA4A, PCED1B, S100A9, TMEM176B0), two with DOG[2] (DOCK2, TMEM176B), and two were unique to that cohort, EMB, and C11orf87. There were 84 DEGs common to all three datasets (DOG[2], TARGET (all patients), TARGET (non-metastatic patients)), 47 (56%) of which

were not in GS-1 or GS-2. Lists of all DEGs and pathways can be found in Supplementary Materials datafile 2.

IPA analysis of the 84 DEGs shared between the canine and human datasets revealed several common immune related pathways of interest, notably TH1 and TH2 signaling, crosstalk between dendritic cells and natural killer cells, NFκβ signaling, immunogenic cell death signaling pathway, and IL-8 signaling (Supplementary Fig. 6a). In addition to the 84 shared DEGs, many of the DEGs that were unique to either the DOG[2] or TARGET datasets were enriched for the same IPA pathways (Supplementary Fig. 6b) including PD-1 - PD-L1 cancer immunotherapy pathway, the IL-10 signaling pathway, and IL-12 signaling and production in macrophages. Significant pathways in DOG[2] but not in TARGET include the osteoarthritis

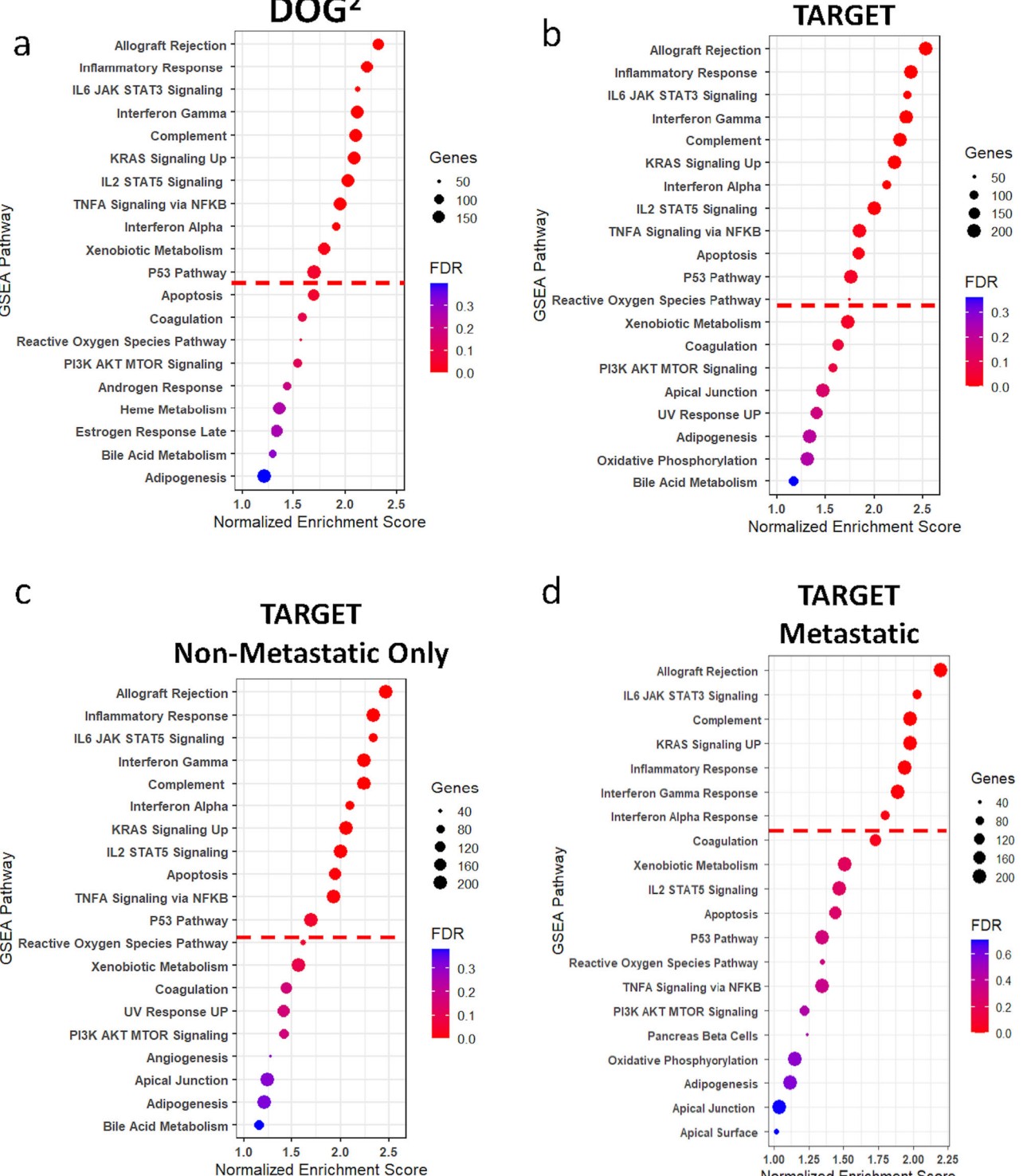

**Fig. 4 Transcriptionally-defined clusters are enriched for specific cellular processes.** Normalized enrichment scores for gene set enrichment analysis (GSEA) of the top 20 pathways over-represented in the favorable prognosis group when compared to the poor prognosis group when clustered by GS-1 in (**a**) DOG[2], (**b**) TARGET, (**c**) TARGET Non-metastatic patients, and (**d**) TARGET metastatic patients. Dot size is representative of the number of genes in the pathway and color is indicative of FDR-q value calculated by software, red line is indicative of significance cutoff. Similar plots for GS-2 and pathways over- represented in the poor prognosis group can be found in Supplemental Figs. 3–5.

pathway and the PI3K/AKT pathway, which has previously been implicated in osteosarcoma[35]. Significant pathways in TARGET but not in DOG[2] include IL-17 signaling, T-cell exhaustion, T-cell receptor signaling, CD28 signaling in T helper cells, allograft rejection signaling, and alternative activation of macrophages pathway.

**Distinct differences in macrophages, B cells, and T cells define cluster populations**. CIBERSORTx was used to infer the cellular composition of the immune infiltrates within tumor tissues, using the bulk mRNA sequencing data (Fig. 5, Table 1). The most notable similarity between the DOG[2] and the TARGET dataset

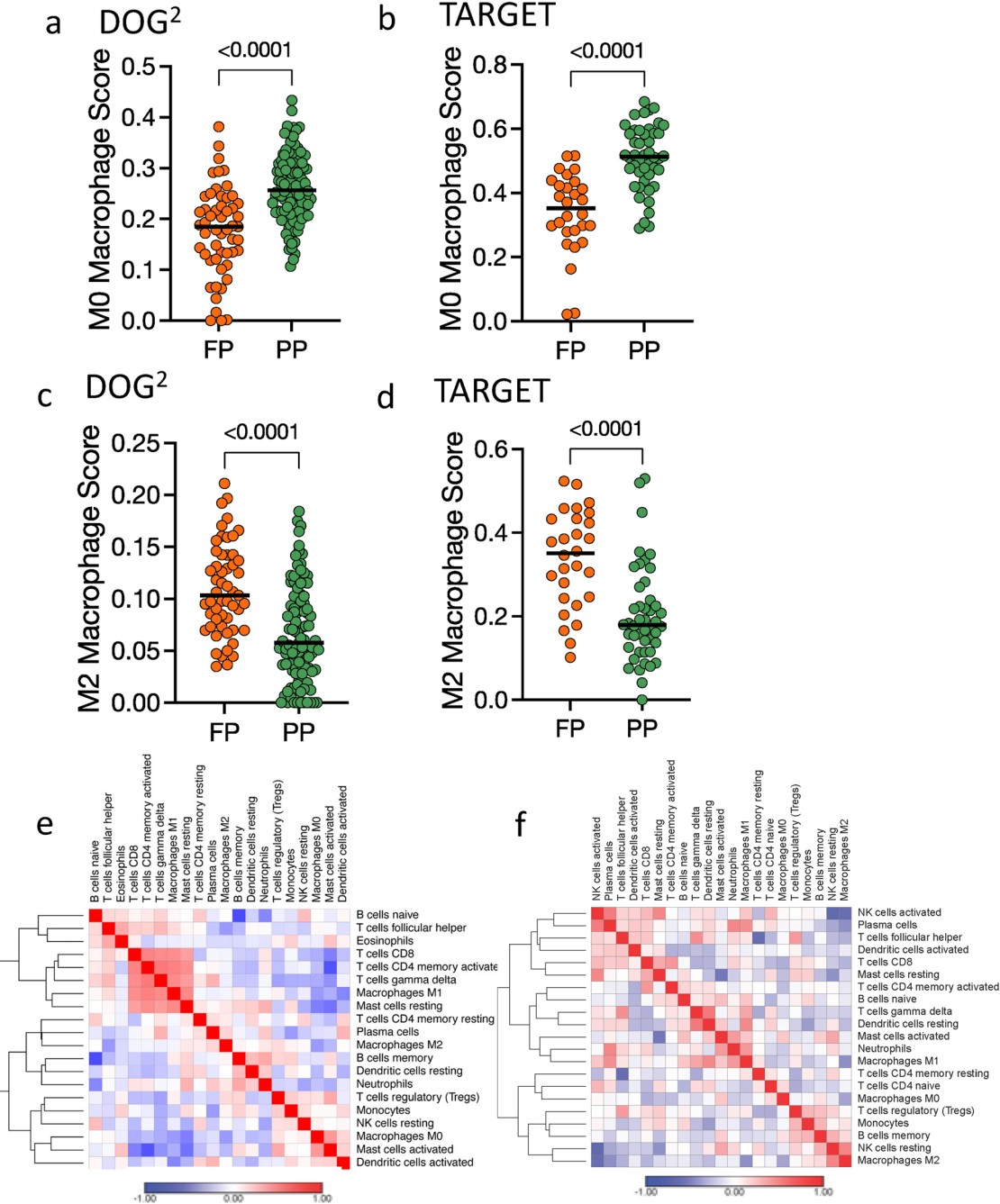

**Fig. 5 Osteosarcoma sub-populations are defined by distinct immune cell populations.** M0 Macrophage CIBERSORTx scores are significantly higher in the poor prognosis (PP) group compared to the favorable prognosis (FP) group in (**a**) DOG$^2$ and (**b**) TARGET. This is the opposite of what is observed for M2 macrophages where higher CIBERSORTx scores are consistently seen in the FP group in (**c**) DOG$^2$ and (**d**) TARGET. Spearman correlations between CIBERSORTx scores in the FP group for DOG$^2$ (**e**) suggest an adaptive immune response characterized by highly coordinated populations of T-cells. Likewise, the same can be seen for cytotoxic lymphocyte response (activated NK and dendritic cells, T-helper cells, and CD8 T cells) in the FP TARGET group (**f**).

occurs for M0 and M2 macrophages. In both datasets, M0 macrophage CIBERSORT scores are significantly higher in the PP group than in the FP group (Fig. 5a, b). This contrasts with the M2 macrophage scores where a significantly higher score is seen in the FP group when compared to the PP group (Fig. 5c, d). Additionally, CD8+ T cells, naïve CD4+ T cells, follicular helper T-cells, and M1 macrophages have significantly higher scores in the FP group and lower scores in the FP group within both DOG$^2$ and TARGET datasets. Furthermore, within the DOG$^2$ FP cohort, we see a high correlation between M1 macrophages, CD8+ T cells, activated memory CD4+ T cells, and gamma delta T-cells (Fig. 5e) which is not present in the PP group (Supplementary Fig. 7a), emblematic of an anti-tumor adaptive immune response primed by activated memory T-cells and orchestrated through M1 macrophages and highly cytotoxic CD8+ and gamma delta T-cells. Similarly, within the TARGET cohort, the correlation between NK cells, follicular helper T-cells, CD8+ T cells, and activated dendritic cells (Fig. 5f) suggests a strong cytotoxic lymphocyte response in the FP group, which is not present in the PP group (Supplementary Fig. 7b).

**Table 1 Mean CIBERORTx scores, expressed as % of cells, between the favorable (FP) and poor (PP) response groups for the TARGET and DOG$^2$ datasets.**

|  | Mean FP TARGET | Mean FP DOG$^2$ | Mean PP TARGET | Mean PP DOG$^2$ | Adjusted P-Value TARGET | Adjusted P-Value DOG$^2$ |
|---|---|---|---|---|---|---|
| B cells naïve | 0.65% | 0.72% | 1.47% | 2.22% | 0.269426 | 0.001869 |
| B cells memory | 0.034% | 1.55% | 0.16% | 0.79% | 0.132465 | 0.000913 |
| Plasma cells | 0.27% | 1.50% | 0.56% | 3.00% | 0.045099 | 0.000105 |
| T cells CD8 | 4.36% | 8.66% | 0.79% | 2.88% | 0.011562 | 2.66E-08 |
| T cells CD4 naive | 0.19% | 0% | 0.59% | 0.16% | 0.045099 | 0.005408 |
| T cells CD4 memory resting | 9.33% | 1.73% | 10.6% | 1.82% | 0.392358 | 0.238212 |
| T cells CD4 memory activated | 1.26% | 4.75% | 1.81% | 1.25% | 0.130614 | 4.62E-06 |
| T cells follicular helper | 1.18% | 0.17% | 0.65% | 0.37% | 0.045099 | 0.048171 |
| T cells regulatory | 1.69% | 2.78% | 0.91% | 3.37% | 0.071562 | 0.079914 |
| T cells gamma delta | 1.44% | 2.14% | 1.12% | 0.26% | 0.317411 | 0.038481 |
| NK cells resting | 0.35% | 17.10% | 1.16% | 17.4% | 0.177486 | 0.318437 |
| NK cells activated | 2.36% | 0% | 1.75% | 0% | 0.13013 | 1 |
| Monocytes | 0.40% | 0.64% | 0.24% | 0.89% | 0.071562 | 0.257392 |
| Macrophages M0 | 33.74% | 17.65% | 51.01% | 26.06% | 1.20E-06 | 6.63E-08 |
| Macrophages M1 | 4.60% | 0.19% | 1.58% | 0.0015% | 6.69E-05 | 4.97E-06 |
| Macrophages M2 | 33.68% | 10.91% | 20.18% | 6.65% | 6.69E-05 | 1.47E-06 |
| Dendritic cells resting | 0.65% | 2.14% | 0.39% | 1.56% | 0.045099 | 0.444684 |
| Dendritic cells activated | 0.11% | 1.48% | 0.16% | 1.62% | 0.317411 | 0.361528 |
| Mast cells resting | 3.38% | 1.05% | 4.59% | 0.077% | 0.071562 | 0.009154 |
| Mast cells activated | 0.21% | 23.85% | 0.14% | 29.01% | 0.317411 | 0.005901 |
| Eosinophils | 0% | 0.012% | 0.013% | 0.0013% | 0.276134 | 0.35374 |
| Neutrophils | 0.12% | 1.02% | 1.17% | 0.56% | 0.119924 | 0.043345 |

The Benjamini Hochberg adjusted p-value indicates significance between the FP and PP groups in the mean CIBERSORTx score.

**Immunohistochemical analysis suggests gene signature as surrogate for immune infiltrate**. To determine whether relative immune cell abundance could be confirmed in canine OS tissues, specific immune cell types were labeled using immunohistochemistry (Supplementary Tables 1, 2). A comparison of the immunohistochemical analysis to the GS-1 signature can be seen in Fig. 6a. In total, the immunohistochemical review of canine OS (Fig. 6b) confirmed the presence of several immune cell types in the majority of OS assigned to the FP group including T cells (CD3), B cells (CD20, MUM1), and macrophages (CD204, Iba1). Furthermore, a comparison between representative samples of the FP and PP cohorts revealed that the FP group was significantly more enriched for CD3, CD20, CD204, CD3, CD45RA, Iba1, and MUM 1 labeling (Fig. 6b). Collectively, these findings suggest that immune-enriched tumors from the DOG$^2$ canine OS cohort are concurrently infiltrated by multiple immune cell subtypes. Finally, five human OS tumor samples labeled with CD3, CD20, and CD204 demonstrate infiltration of human OS by T cells, B cells, and macrophages, respectively (Fig. 6c). As seen in canine OS tissues, CD204+ cells are numerous and are observed throughout the human OS tissue.

**Additional analyses reveal novel transcriptionally-defined subtypes**. The signatures GS-1 and GS-2 taken from Gardner et al. and assessed herein provided evidence that a predictive gene signature could be independently found in canine osteosarcoma data and additionally be equally predictive in comparable human osteosarcoma data. This motivated us to ask, if first, we could independently derive predictive signatures directly from the DOG$^2$ dataset that could be applied to the TARGET dataset and, second, if these signatures could outperform GS-1/GS-2, particularly in the non-metastatic TARGET cohort. The comparative analysis of DOG$^2$ canine data, which is entirely derived from canine patients free of macroscopic metastasis at diagnosis, is most closely clinically related and has the most translational

relevance to non-metastatic human patients represented in TARGET. Further, this is the patient population for which therapeutic stratification and additional prognostic biomarkers are needed to improve beyond neoadjuvant chemotherapy-induced % necrosis in humans, which does not reliably identify all patients at high risk for metastatic progression[36–38] Single Sample Gene Set Enrichment Analysis using the 52 hallmark gene sets from the mSigDB on DOG$^2$ followed by k-means clustering identified two distinct clusters (Fig. 1b). Kaplan–Meier analysis yielded curves for each cluster with significantly different DFI and OSv (Supplementary Fig. 8a, b) for these canine patients. Differential gene expression analysis between the two groups identified 259 differentially expressed genes. Interestingly, 19 of those genes overlapped with GS-1 and 61 overlapped with GS-2, again highlighting the relevance of these previously identified signatures within the DOG$^2$ dataset. Additionally, IPA analysis of the DEGs returned 45 significant pathways (Supplementary Fig. 8c). Of these, 22 pathways are related to inflammation or immune function and 36 overlapped with the IPA analysis of DEGs associated with GS-1. The large overlap in pathways is not surprising since 173 genes are also in the DEG analysis between FP and PP groups, underscoring the underlying immune-focused theme in the DOG$^2$ data that is associated with prognosis.

Given that 259 genes from our signatures, including those in GS-1 and GS-2, comprise multiple immune processes, we sought to identify granular subsets of genes and samples in the data by employing bi-clustering using iterative search algorithm (ISA, approach outlined in Fig. 1b). The ISA algorithm yielded 4 bi-clusters covering a total of 103 genes (ISA 1: 30, ISA 2: 19, ISA 3: 39, ISA 4: 15) (Supplementary Table 3). While a definitive pattern of overexpression can be easily observed for each bi-cluster over the four ISA-derived gene signatures (Fig. 7a–d), only ISA sample bi-cluster 4 yielded a significant difference in DFI in canines upon Kaplan–Meier analysis (Fig. 7h). In addition, although not meeting a strict significance criterion of $p < 0.05$, ISA sample bi-cluster 3 shows noticeable difference in DFI at earlier time points

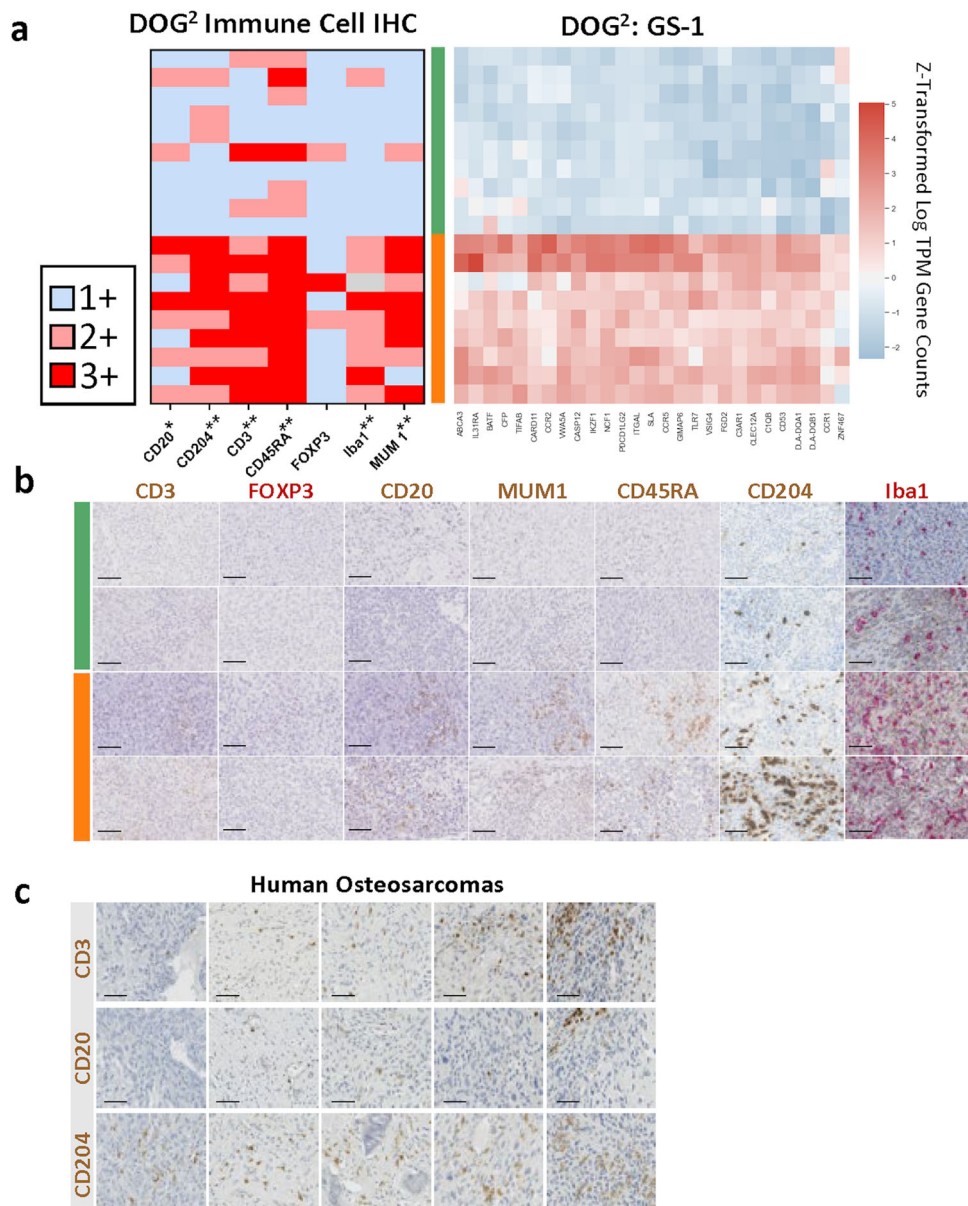

**Fig. 6 Immunohistochemical staining supports use of gene signatures as surrogate for immune infiltrates in canine osteosarcoma.** For immunohistochemical (IHC) analysis, a subset of cases from the GS-1 Poor Prognosis (Green) and Favorable Prognosis (Orange) clusters were selected. **a** Based on IHC labeling of antibodies listed along the x-axis, cases were categorized using a 3 point scale as immune high (3+), immune intermediate (2+), or immune low (1+). Iba1 was not scored in one sample (identified in gray) due to poor tissue sectioning. All other tissues were included as indicated. The adjacent heatmap shows gene expression for the IHC cases based on GS-1 hierarchical clustering of all samples and illustrates the relationship between the GS-1 clusters and IHC category. Asterisks indicate significant differences between clusters in IHC quantification with *$p < 0.05$ and **$p < 0.005$. **b** Example images of the IHC labeling from the Poor Prognosis (Green) and Favorable Prognosis clusters (Orange). **c** Example images of IHC labeling for CD3, CD20, and CD204 in human OS samples. Scale bar = 50 µm.*IHC labels for Figs. B and C represent IHC chromogen (either red or brown) used to label positive cells.

that results in a 14-week difference in median survival (Fig. 7g). Given the rapid progression to macroscopic metastatic disease that often occurs in the canine within the first 6 months after diagnosis with the current standard of care, this could carry significant clinical importance for veterinarians.

We wanted to further determine if these new signatures had any relevance in the TARGET data and specifically in the non-metastatic cohort, which is the closest human patient comparator to the canine OS patients that comprise DOG². We applied a similar strategy as with the GS-1 and GS-2 signatures, using K-means clustering to cluster the samples based on the gene expression data. Based on this clustering, Kaplan–Meier analysis was performed between clusters to ascertain if the given gene signature was associated with a significant difference in patient outcomes. Three of the signatures produced two distinct clusters: ISA gene signature 1, ISA gene signature 2, and ISA gene signature 3 in both the entire TARGET dataset (Fig. 8) and the non-metastatic TARGET patient subset (Fig. 9). Interestingly, despite not showing any prognostic value in canine, ISA gene signature 1 was prognostic for progression free survival in the TARGET data, specifically only in the non-metastatic patients (Fig. 9d). Likewise, ISA signature 3 was significantly prognostic in both the entire TARGET dataset (Fig. 8f) and non-metastatic patients (Fig. 9f). ISA gene signature 4 produced unstable clusters, the clusters and the differences between Kaplan-Meier curves would

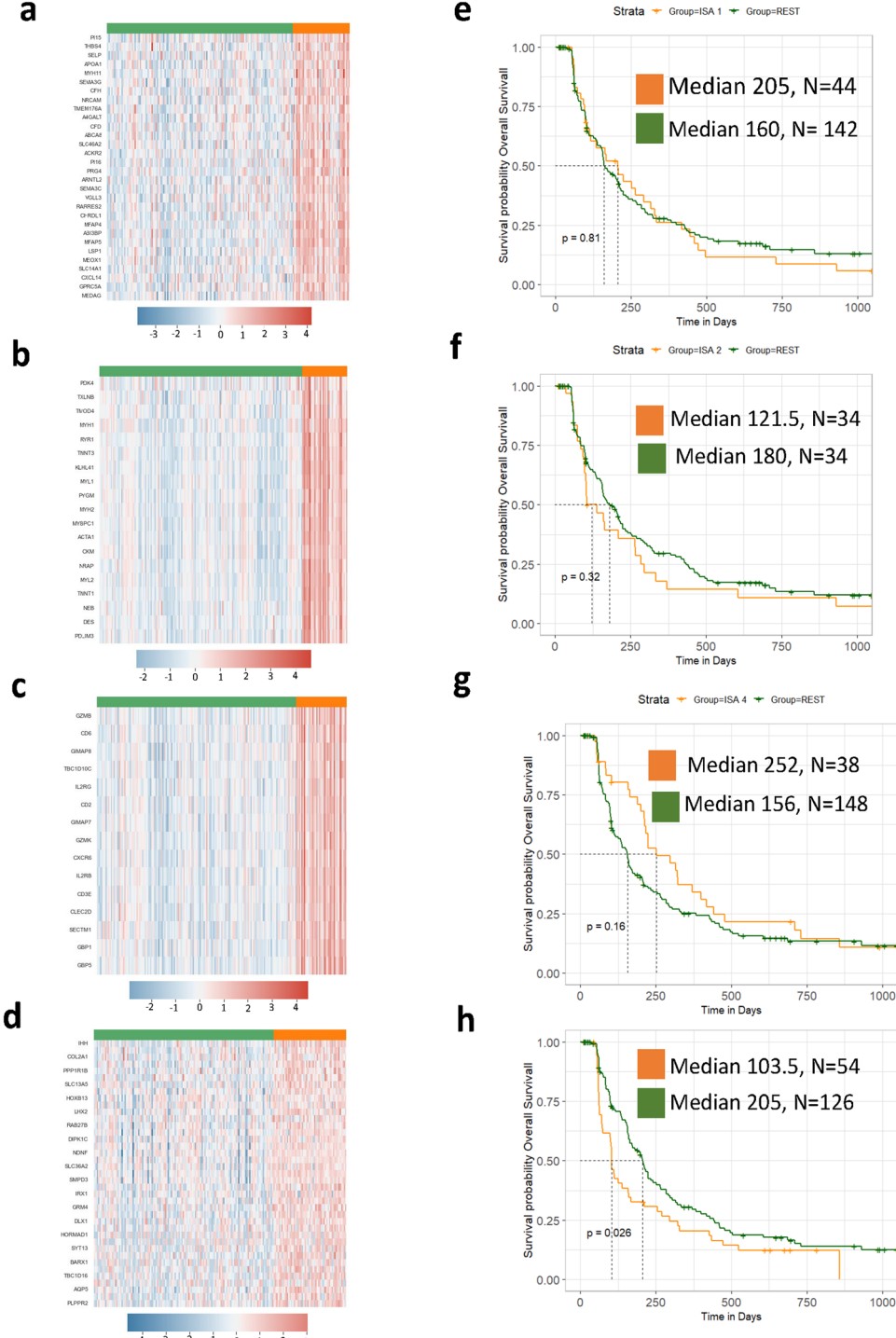

**Fig. 7 Bi-clustering defines additional gene signatures and sub-populations in canine osteosarcoma.** DOG² gene expression profiles for (**a**) ISA signature 1, (**b**) ISA signature 2, (**c**) ISA signature 3, and (**d**) ISA signature 4. Corresponding Kaplan–Meier curves of ISA sample bi-cluster versus all other samples for ISA signature 1 (**e**), ISA sample bi-cluster 2 (**f**), ISA signature 3 (**g**), ISA signature 4 (**h**). Y-axis is Overall Survival, with medians reported in days from diagnosis.

change drastically based on the inherent stochastic nature of the *K*-Means algorithm. However, Cox regression analysis on genes in ISA signature 4 revealed that Iroquois homeobox 1 (IRX1) was prognostic of progression free survival with higher expression ultimately leading to poor outcomes (Supplementary Fig. 9a), which has been previously reported in an unrelated human OS dataset as well as predicting lung metastasis in murine models[39]. This behavior is mirrored in the canine (Supplementary Fig. 9b).

An essential question to this analysis is whether these signatures have any biological significance that points to differences among the various populations of patients and/or tumor cells and associated microenvironments. ISA gene signature 3 was most associated with immune processes including T-cell binding and response (CD2, CD3E, CD6, CXCR6, CLEC2D[40], GZMB), Interferon Gamma Response (CLEC2D[41], GBP1, GBP5), Natural Killer Cells (CLEC2D[40], GZMB, GZMK),

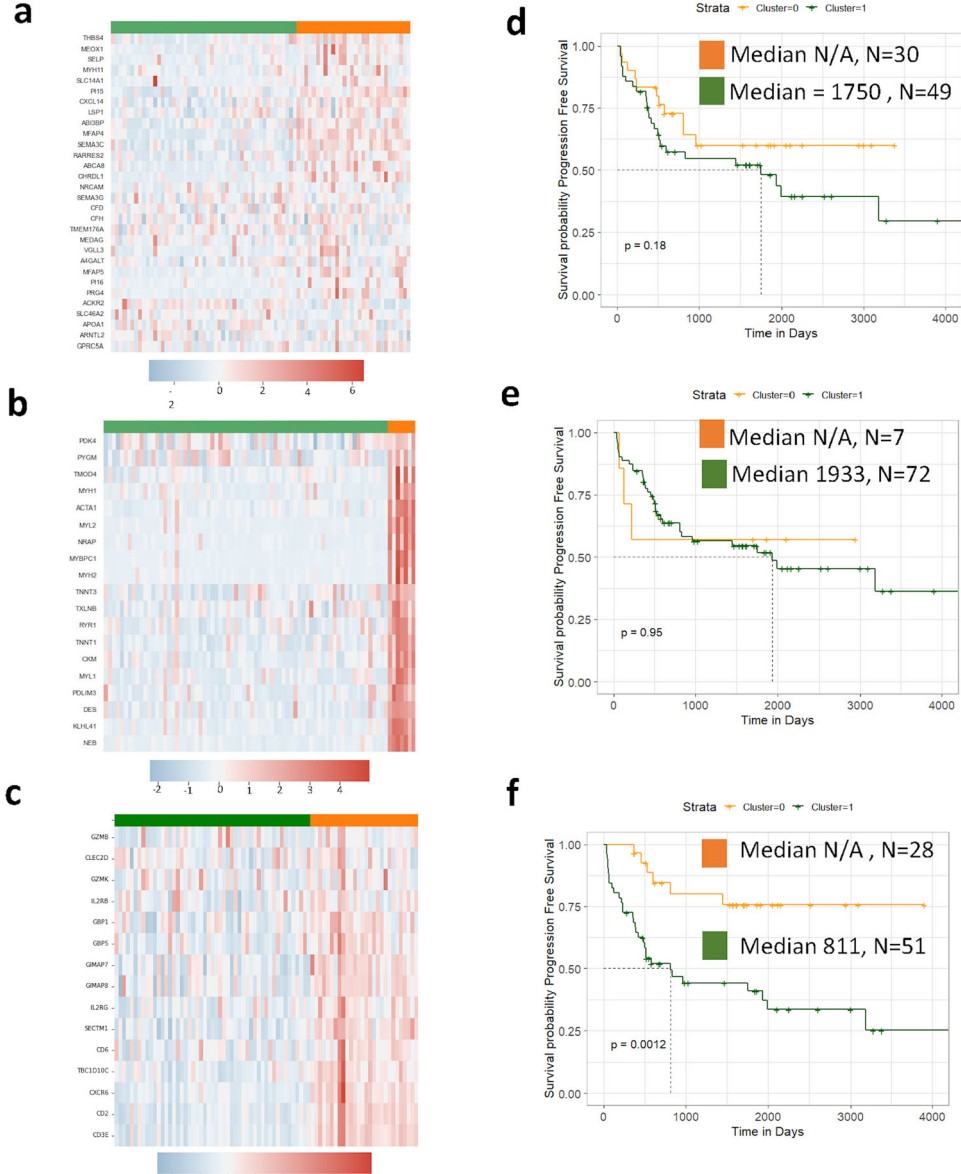

**Fig. 8 Novel canine ISA-derived gene signatures are prognostic in human osteosarcoma.** TARGET gene expression profiles for (**a**) ISA gene signature 1, (**b**) ISA gene Signature 2, and (**c**) ISA signature 3. Corresponding Kaplan–Curves for *K*-Means clusters formed from (**d**). ISA gene signature 1, (**e**) ISA gene signature 2, and (**f**) ISA gene signature 3. *ISA gene signature 4 was omitted as it did not form consistent clusters. Y axis is Progression Free Survival. Medians are reported in days from diagnosis.

and IL2RB and IL2RG which are involved in many processes including but not limited to IL-2 signaling, IL-15 signaling, and IL-9 signaling. Several genes in ISA gene signature 1 relate to neutrophil motility (LSP1[42], CXCL14[43], SELP[44]), dendritic cell regulation (ACKR2[45], TMEM167A[46], CXCL14[47]), macrophage regulation (ACKR2[45], CFH[48]), inhibition of tumor cell migration (SEMA3G[49]), and lung tumor suppressor (GPRC5A[50]). Despite ISA gene signature 2 not being prognostic, it clearly exhibited expression patterns that could be seen in both humans and canines. In addition, many genes in ISA gene signature 2 have been associated with the DMD gene in mice including MYL1 and MYH2[51], DES[52], MYBPC1, MYH1, TMOD4, PYGM, and PDK4[53]. This might be of particular importance in the same manuscript from which GS-1 and GS-2 were derived. In this work, Gardner et al. identified mutations in dystrophin (DMD) in 2 of the 24 dogs analyzed by WGS[21]. ISA gene signature 4 does not necessarily present with any particularly cohesive theme

despite being the largest signature. It contains two collagen genes (COL11A2, COL2A1) several solute carrier genes (SLC8A3, SLC13A5, SLC36A2) which have been shown to be prognostic in osteosarcoma[54], and genes involved in Wnt signaling (APCDD1L, BARX1) which might be significant given the oncogenic role Wnt signaling plays in osteosarcoma[55,56].

## Discussion

The 5-year survival rate for humans with localized OS stands stagnant at approximately 70%, underscoring the need for informative animal models to assist with novel therapeutic discovery and development efforts[1–3,57,58]. Similarly, despite the use of multimodal therapy, the prognosis for canine OS is uniformly poor, with median survival time of 5 to 13 months for patients presenting with localized disease[59,60]. Some of these dogs will be enrolled in clinical trials aimed at improving outcomes in canine

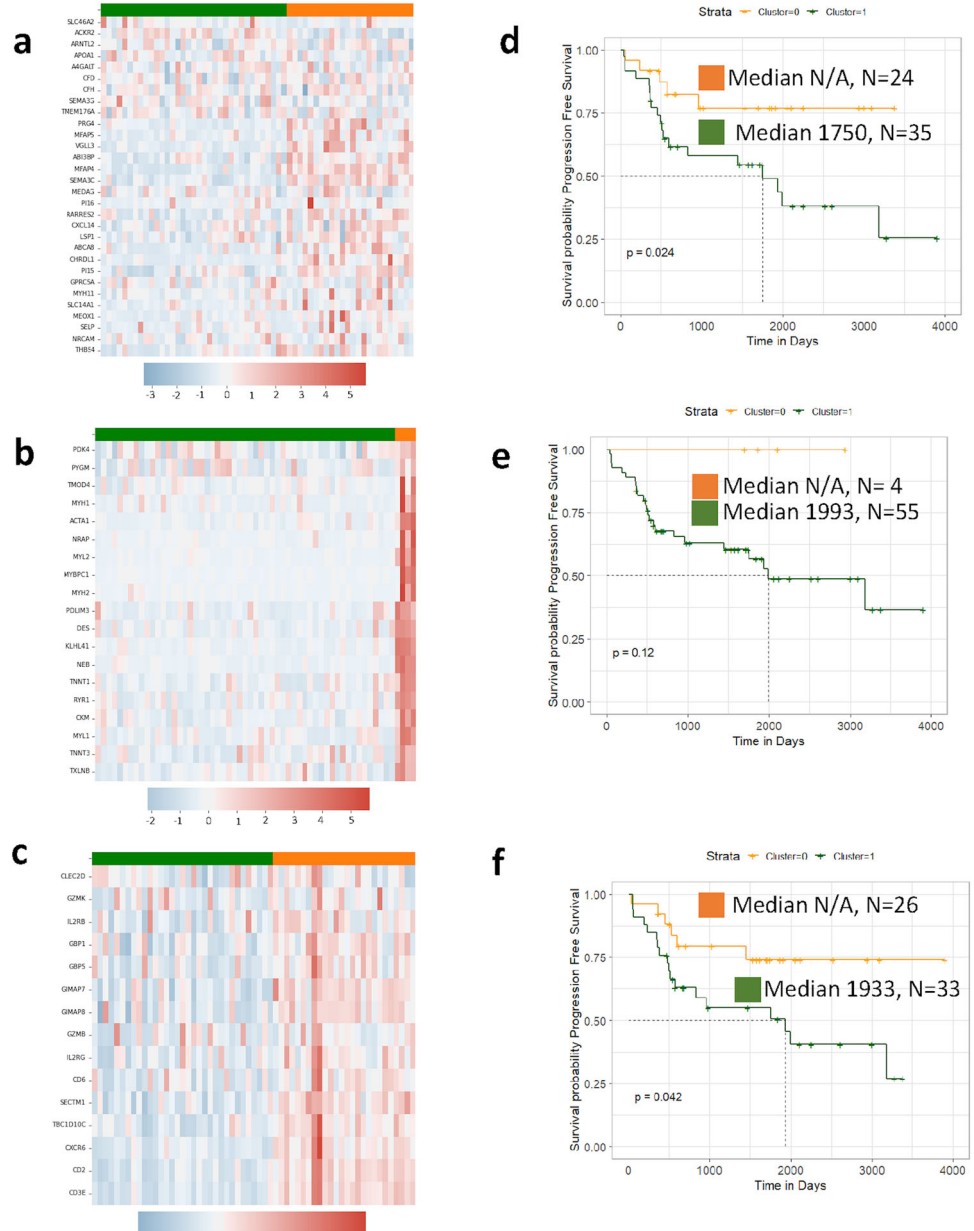

**Fig. 9 Novel canine ISA-derived gene signature are prognostic for progression free survival in human osteosarcoma in patients without metastatic disease at the time of diagnosis.** TARGET (non-metastatic patients) gene expression profiles for (**a**) ISA gene signature 1, (**b**) ISA gene Signature 2, and (**c**) ISA signature 3. Corresponding Kaplan–Curves for *K*-Means clusters formed from (**d**) ISA gene signature 1, (**e**) ISA gene signature 2, and (**f**) ISA gene signature 3. *ISA gene signature 4 was omitted as it did not form consistent clusters. Y axis is Progression Free Survival. Medians are reported in days from diagnosis.

and human OS patients. One of the main challenges for OS, as with most cancers, is determining whether treatment and outcome may be informed by identifying sub-populations of patients with different underlying tumor biology and potential unique sensitivities to specific therapies. Previous studies have highlighted prognostic factors such as histological subtype[61], histological grade[62], tumor location[63], and serum alkaline phosphatase levels[62,64].

Here we show that transcriptional signatures derived completely in canine osteosarcoma, both previously reported as well as those defined de novo through a ssGSEA and ISA approach, divide both canine and human data into two distinct populations. When all TARGET data is considered, both progression free survival and overall survival is significantly associated with the cluster assignment determined by these gene signatures.

However, the presence of metastatic disease eclipses any predictive capacity for either gene signature applied herein. This is not surprising given the dismal prognosis for advanced stage disease and the refractory nature of macroscopic metastasis to therapy. The larger sample size and uniform nature of the DOG[2] cohort may also improve predictive signature discovery compared to TARGET – which encompasses human patients that likely underwent a much wider variety of therapies and non-standardized disease monitoring protocols. However, from the standpoint of survival analyses, the issue of humane euthanasia in dogs makes assessment and comparative evaluation of overall survival between dogs and humans problematic as owners' must decide whether to pursue additional therapy vs. euthanasia when faced with progression of their pet dog's disease. One way to

improve the translation of canine clinical trials is to assess disease free interval (DFI) as the primary clinical endpoint, which is an invaluable facet of the DOG[2] dataset. Critically, as stated previously, metastatic progression is of vital relevance to both species since the development of metastases remains the major prognostic factor in human and canine patients.

If we use DOG[2] as a distributional model, then based on the number of human metastatic samples in the TARGET dataset (n = 20), the expected number in the FP group would be 6.3 and the expected number in PP group is 13.7. Comparing this to the real values given by GS-1 of 5 and 15 this would yield a similar distribution of patients in the FP and PP groups as tested by a chi square goodness of fit test ($\chi^2 = 0.391$, $p = 0.531$). Collectively this analysis provides evidence that there is a population of human OS in which these gene signatures might be prognostic as they are in canines. Further, the transcriptional profiles are conserved across both DOG[2] and TARGET data strongly suggesting an underlying common biology. The comparative approach provides significant advantages for the discovery of predictive factors that may be applicable to OS, bolstered by the commonality of the disease in the pet dog population, the high rate of metastasis, and the ability to conduct large-scale canine OS clinical trials with concomitant collection of biologic specimens[65]. If a larger cohort of prospectively collected human data with harmonized treatment and follow up measures was available, such as from a clinical trial as in the DOG[2] cohort, the approach here may have more predictive power for humans. However, the inherent heterogeneity of the disease and differences between canine and human tumor biology must also still be considered in this context.

Additional transcriptional analysis such as GSEA, CIBER-SORTx, and differential expression, supported by immunohistochemical analysis in DOG[2], suggests in both DOG[2] and TARGET the importance of immune processes and immune cells in disease progression and outcome. GSEA revealed enrichment in the FP group of several pathways involved in innate immunity and inflammation such as interferon response and complement system. This is supported by the work of Scott et al. and Wan et al. who have previously shown that immune based gene signature equated to better prognosis in humans[66,67]. Additionally, according to estimates acquired using CIBERSORTx, our results suggest that differences in macrophage, plasma cells, and T-cell populations possibly influence disease progression and survival in both canine and human OS. However, a notable difference between DOG[2] and TARGET was observed in B cell populations namely an increase in B cells, supported by immunohistochemistry, in the FP group of DOG[2] that was not observed in TARGET.

Although computational algorithms such as CIBERSORT and CIBERSORTx have been used in canine studies[25,68,69], it is important to note that these algorithms have largely been developed and tested with respect to human data to deconvolute bulk mRNAseq data into relative representations of cellular populations. Although canine and human cells are highly comparable with respect to cellular markers, there are some differences, such as CD4 positivity of canine neutrophils, that could affect such computational analysis[70,71]. To this point, CIBER-SORTx estimated a high proportion of mast cells within our canine osteosarcoma tissues. However, we were unable to confirm the presence of mast cells using toluidine blue (Supplementary Fig. 10), a histochemical staining technique that has been used to quantify mast cells in decalcified bone lesions in other studies[72–75]. Interestingly, mast cells have been described in human OS samples[76] and made up a significant proportion of cells in our CIBERSORTx results obtained from human TARGET data (4.5%, the fourth most abundant cell type). Osteosarcomas are heterogeneous tumors thus regional differences in the tumor tissue submitted for RNAseq and the FFPE block can occur; nonetheless, our data suggest that further work is needed to delineate the potential roles of mast cells in the context of osteosarcoma and to validate how the combination of the LM22 signature and the CIBERSORTx algorithm behaves specifically within canine osteosarcomas that develop alongside a naturally co-evolving immune microenvironment.

The character and robustness of the immune cell response are known to play a critical role in canine cancer. For example, dogs with OS are reported to have significantly fewer circulating Tregs while a decreased CD8+ / Treg ratio is associated with poor survival[77]. Likewise, in human OS, higher levels of CD8+ T-Cells have been associated with better outcomes[78–80] and are subject of new immunotherapies[81]. Furthermore, in an immunohistochemical analysis of 30 canine OS tumors, a higher number of CD204-positive cells was associated with a significantly longer DFI[82]. Additionally, Das et al. reported that 22 immune related gene-sets were enriched in dogs with longer DFIs in a small sample set of 26 dogs[25]. These and the present study underscore the importance of the immune response in cancer progression.

The tumor microenvironment is host to several immune cells acting in concert, among these are the tumor associated macrophages (TAMs) and they have gained particular interest within the context of OS[83,84]. The data we present certainly points to macrophage biology that is common to both canine and human OS tumors. For instance, with respect to the CIBERSORT analysis, M0 and M2 macrophages make up a significant proportion of the cell population in both DOG[2] and TARGET. They also follow the same pattern in which M2 macrophages are more abundant in the FP group and more M0 macrophages are seen in the PP group. Additionally, with respect to M1 macrophages, while they are present in a much smaller fraction, the fraction in the FP group is much greater: 13x for DOG2 and 3x for TAR-GET. In addition to the elimination of tumor cells through phagocytosis, M1 macrophages are thought to stimulate and enhance the cytotoxicity of other leukocytes through increased tumor antigen presentation and promotion of pro-inflammatory cytokines[85,86]. Alternatively, M2 macrophages are associated with immune suppression, decreased antigen presentation, growth factor release, and promotion of angiogenesis; all thought to be conducive to cancer progression[86,87]. However, in OS, there has been mixed evidence as to how TAMs influence overall patient outcomes. Much of the literature concerning the relationship between TAMs and OS suggests a correlation between overall macrophage infiltration, consisting of both M1 and M2 phenotypes, and increased progression free interval or overall survival. Deng et al. used the CIBERSORT algorithm to show better overall survival with increased M1 and M2 macrophage infiltration. Similarly using genomic profiling and IHC, Buddingh et al. established in a cohort of 63 patients that total TAM, both M1 and M2 phenotype, were associated with better outcome[88]. An additional study that used CD163 to quantify TAM in 124 human OS tumors found that increased TAMs resulted in longer progression free survival and overall survival[89] which is complemented by a similar canine study that reported longer DFIs with increased TAMs[82].

Our transcriptomic analysis indicates that M2 macrophages are abundant in canine OS, with IHC detection of numerous macrophages expressing CD204, a macrophage scavenger receptor highly expressed in M2 macrophages in humans[90]. M2 macrophages have been associated with poor outcomes in multiple other tumor types[91–93]; however, DOG[2] data suggest that having a greater number of M2 macrophages might be beneficial. The answer might be unique to the bone itself, wherein macrophages and osteoclasts are both derived from monocytes, while IL-4 and

IL-13 needed for the differentiation of a monocyte into M2 macrophage are inhibitors of osteoclast formation[94]. Therefore, it is possible that a shift in monocyte differentiation to favor M2 macrophages reduces osteoclast formation and thereby inhibits bone resorption which might be essential to tumor development. Furthermore, M2 macrophages are known or produce a number of cytokines that actively regulate osteoclasts including IL-10, BMP-2, TGF-β1, OPN, and 1, 25 dihydroxy vitamin $D_3$[95]. There is some evidence that suggests inhibiting osteoclast formation decreases local tumor growth and increases survival. For example, Lamoureux et al. showed in a rodent model that enhanced osteoprotegrin (OPG), which also inhibits osteoclast formation, led to decreased local tumor growth and a 4-fold increase in mouse survival[96]. Additionally, Ohba et al. established that osteosarcoma growth directly correlates with tumor-induced osteolysis and activation of osteoclasts in vivo[97]. Furthermore, the high abundance of CD204-positive macrophages occurred concurrently with elevated B and T cell tumor infiltration which may indicate a higher pan-immune cell response is associated with improved prognosis, regardless of the macrophage subtype. The importance of an effective immune response is also supported by the relative abundance of immune cell types in canine OS, namely that 3 of the 4 cell types with higher relative abundance in the PP group (GS-1 signature) are naïve or non-activated (naïve B-cells, naïve CD4 T, M0 macrophages) which may indicate an inadequate anti-tumor immune response.

We have shown that many of the genes put forth in Gardner et al. can be independently derived directly from DOG² data using ssGSEA and ISA-based bi-clustering, and these genes reflect a difference in immune processes between two distinct patient populations. However, as we generally characterize the difference between groups as an immune hot and immune cold, this dichotomy incompletely represents the spectrum of patient samples. Using different clustering methods led to four new gene signatures with foundations that suggest relevant biological differences. Notably, unlike Gardner's signatures, two of the signatures were prognostic in the non-metastatic cohort TARGET data. However, these signatures were not significantly prognostic in the DOG² data. This might be best reconciled in ISA bi-cluster and gene signature 3 which shows a clear deviation at early timepoints in the DOG² data demonstrating the ability of gene signatures to detect disease progression with early metastatic failure in canine OS. Considering that ISA gene signature 4 is composed of several genes related to T-cell processes, it is possible that patients with this signature can illicit an anti-tumor immune response, but as time progresses, processes such as T-cell exhaustion become more dominant. Analysis of longitudinal datasets collected from the same patient could help support this hypothesis. The dependence on T-cells might explain why this signature performs better in non-metastatic TARGET samples than GS-1 or GS-2 as greater T-cell activity might discourage the development of metastasis. While ISA gene signature 2 does not have the prognostic capacity, it does point to a possible common population in human OS and canine OS with DMD mutations, which will be investigated through comparative analysis of matched whole-genome sequencing datasets. In terms of ISA gene signature 4, we show agreement with earlier findings[39] that overexpression of IRX1 leads to poor prognosis both in humans and canines. ISA gene signature 1 was derived from our canine patients and identified a similar expression pattern in both the canine and human datasets; however, while it is prognostic in human OS, it fails to be in canine. This supports the potential for canine tumor-derived signatures to yield clinical relevance for human patients. The lack of prognostic significance in our canine samples may reflect the degree of genomic and transcriptomic complexity of the disease or the disparate treatment approach in

dogs vs. humans that may influence the ability to identify gene signatures with prognostic capacity in both species. For example, canine patients that comprise the DOG² cohort have undergone limb amputation and up to 4 cycles of carboplatin chemotherapy, with very few dogs undergoing additional treatment after the detection of metastatic progression. This is in stark contrast to humans with OS, who often undergo surgical metastasectomy as well as repeated rounds of salvage chemotherapy if disease progression occurs. Further, euthanasia is the most common cause of death in dogs and is often a reflection of the owners' wishes rather than the extent of the disease. This may be further compounded by a shorter lifespan which can preclude the identification of significant differences in OS between populations. For these reasons, in the comparative oncology context, emphasis should be placed on identifying factors that have predictive value for disease-free intervals rather than overall survival.

In this study, we have provided evidence that distinct differences in immune cell populations result in significant differences in outcomes in canine OS patients. Applying the same analysis to the mRNA-Seq portion of the TARGET dataset we were able to provide strong evidence to establish that these transcriptomic and clinical patterns identified in canine OS patients could be applied comparably to human OS. Further analysis revealed shared cellular processes in both species, suggesting a transcriptional program that exhibits a rich interplay between tumor cells and immune cells in the tumor microenvironment. This provides a springboard for further investigations of comparative single-nuclei transcriptomics and geospatial transcriptional profiling. Further, evidence of shared cellular processes and associated biology between canine and human OS primary tumors underpins the value of the dog as a patient model for human OS and provides opportunities to explore these pathways as druggable targets in both species. Furthermore, the canine patient as a preclinical OS model is bolstered by the higher annual incidence of OS in dogs compared to people. From a purely computational standpoint, increased volume and availability of canine -omics data could allow computational scientists to discover new therapeutic targets and treatment strategies that could then be directly applied to human OS. Canine clinical trials and the valuable data they provide will be extremely beneficial to researchers and clinicians, both in the veterinary and medical communities, by improving our understanding of the disease and advancing the development of effective treatments. Such foundational groundwork is an essential aspect in determining how future oncology clinical trials assessed in pet dogs can best be leveraged to benefit human OS patients.

## Methods

Demographic and clinical outcome data, inclusive of canine patient demographics, tumor location, and serum alkaline phosphatase (ALP) status was curated from the canine clinical trial patient cohorts enrolled in the National Cancer Institute's Comparative Oncology Trials Consortium (COTC) 021/022 clinical trials[65]. (Supplementary Tables 4, 5). Privately-owned pet dogs, which varied in age, breed, and sex, were enrolled at participating COTC veterinary academic institutions, with each institution maintaining compliance with its own Institutional Animal Care and Use protocol for the clinical trial, which contained uniform trial procedures and a harmonized clinical care/monitoring schedule. This cohort of canine patients is henceforth referred to as DOG² (Decoding the Osteosarcoma Genome of the Dog) and consists of 324 dogs[65]. All tumor biopsies were obtained prior to treatment and were evaluated by anatomic veterinary pathologists at participating COTC institutions (https://ccr.cancer.gov/comparative-oncology-program/consortium). After tumor sample QA/QC and medical record review, 186 primary tumors were moved forward for analysis. All dogs had a confirmed diagnosis of appendicular osteosarcoma and no evidence of macroscopic metastatic disease based on standard staging methods and no prior treatment. All dogs underwent amputation of the affected limb and subsequent randomization into one of two treatment arms: (1) adjuvant carboplatin alone ($n = 93$; standard of care/SOC) or (2) adjuvant carboplatin plus rapamycin ($N = 93$; SOC + rapamycin). Significant differences in outcome between the two different treatment arms were not observed (Supplementary Fig. 11). The TARGET dataset was accessed through the

National Cancer Institute Genomic Data Commons at https://portal.gdc.cancer.gov/[98]. A total of 88 RNA-SEQ files were downloaded and compiled into a single data matrix. Corresponding clinical data was available for 85 of 88 specimens. The RNA-seq data from 6 specimens were excluded because they had fraction exon and coding bases less than 0.5 and an intergenic rate greater than or equal to 0.3, leaving RNA-seq data from 79 human specimens suitable for downstream analysis.

**Nucleic acid isolation**. RNA was isolated from canine frozen tumor tissue in RNAlater using Qiagen Allprep DNA/RNA Mini Kit (Cat#80204). The total RNA quality and quantity was assessed using Nanodrop 8000 (Thermofisher) and Agilent 4200 Tapestation with RNA Screen Tape (Cat# 5067-5576) and RNA Screen Tape sample Buffer (Cat#5067-5577). All samples forwarded for mRNA sequencing had a RIN > 8 and a total RNA quantity > 100 ng.

**Library Preparation and mRNA sequencing**. Between 100 ng to 1ug of total RNA was used as the input for the RNA sequencing libraries. Libraries were generated using the TruSeq Stranded mRNA library kit (Illumina) according to the manufacturers protocol. The libraries were pooled and sequenced on NovaSeq S1 using a $2 \times 150$ cycle kit. The HiSeq Real Time Analysis software (RTA v.3.4.4) was used for processing raw data files. The Illumina bcl2fastq2.17 was used to demultiplex and convert binary base calls and qualities to fast format. The samples had 44 to 61 million pass filter reads with more than 91% of bases above the quality score of Q30. Reads of the samples were trimmed for adapters and low-quality bases using Cutadapt. The trimmed reads were mapped to the CanFam4 reference genome (GSD_1.0[99] from NCBI) using STAR aligner (version 2.7.0 f) with two-pass alignment option. RSEM (version 1.3.1) was used for gene and transcript quantification based on the CanFam4 GTF file. The average mapping rate of all samples was 83% with unique alignment above 66%. There were 13.13–26.26% unmapped reads. The mapping statistics were calculated using Picard software. The samples had between 0.01–0.76% ribosomal bases. Percent coding bases were between 58–71%. Percent UTR bases were 10–16%, and mRNA bases were between 75–82% for all the samples. Library complexity was measured in terms of unique fragments in the mapped reads using Picard's MarkDuplicate utility. The samples had 48–78% non-duplicate reads. In addition, the gene expression quantification analysis was performed for all samples using STAR/RSEM tools.

**Computational methods**. For the DOG[2] dataset, filtering was performed to remove low count genes using the R function filterByExpr of the edgeR package using the parameters (min.count = 10, min.prop = 0.5, large.$n$ = 5), leaving 13408 of 37952 probes. For the TARGET dataset, genes denoted as protein coding or polymorphic pseudogene were used, leaving 20010 total genes. Each dataset was then normalized using quantile normalization using the R package normalize.quantiles[100]. The gene signatures applied herein were derived from differential expression analysis between canine OS tumors and a canine reference osteoblast as outlined in Gardner et al.[21]. After filtering the DOG[2] dataset, 87% (27 of 31) of the genes in the first signature (GS-1) and 58.1% (157 of 270) of the genes in the second signature (GS-2) remained. Similarly, almost all the genes in GS-1 and GS-2 present in DOG[3] were conserved in TARGET except for 2 canine specific genes. Clustering analyses were performed using packages accessible through Python 3.8.3. (Fig. 1a). K-means clustering was performed using Scikit-learn version 0.23.1. Clustering was done independent of any associated clinical outcomes. The optimal number of clusters in the datasets was determined by ranging $k$ from 2 to 6 and selecting the number of clusters with the maximum silhouette score[101]. Kaplan–Meier curves were constructed for each cluster and compared using the survminer package version 0.4.9 using R version 4.0.3. Gene set enrichment analysis (GSEA)[102] was performed using version 4.1.0. using molecular signatures in the msigDB hallmarks version 7.5[33]. For differential expression analysis raw gene expression was first filtered for low counts by the filterByExpr package using the parameters (min.count = 10, min.prop = 0.5, large.$n$ = 5) the data was then normalized using the trimmed mean of m-values (TMM) method[103] differential genes were then calculated using the Limma-Voom pipeline[104] of the edgeR R package. To determine pathway enrichment, DEGs that met a cutoff criterion of a fold change greater than 3 and an adjusted p-value less than 0.05 were used as input to Qiagen Ingenuity Pathway Analysis (IPA)[105]. Virtual immune cell profiling was carried out using the CIBERSORTx algorithm[106], https://cibersortx.stanford.edu/index.php, on non-log transformed TPM count data using the LM22 leukocyte gene signature from Newman et al.[107]. P-values for the CIBERSORTx results were calculated based on 100 random permutations and samples with a p-value greater the 0.05 were removed from further analysis. The CIBERSORTx scores for each cell type were evaluated between cluster groups using a Mann–Whitney U-test.

We then sought out to devise novel gene signatures beyond what was previously reported in Gardner et al. Single Sample Gene Set Enrichment Analysis (ssGSEA)[108] was performed using version 10.1.0 using Gene Pattern web server as follows[109] (Fig. 1b). Low gene counts were filtered using filterByExpr (min.count = 10, min.prop = 0.5, large. $n$ = 5) and normalized using (TMM)[103] by the calcNormFactors function in the edgeR package and the Log 2 transformed counts per million leaving a data matrix 186 samples by 13408 probes. The few probes that mapped to the same gene were averaged, while several of the

uncharacterized probes were removed because of ambiguity in naming conventions. This left a final data matrix of 186 samples by 13164 genes. Using the hallmark gene signatures[33] in the mSigDB, ssGSEA was performed resulting in a transformed representation of the data. With this transformed data matrix K-Means clustering was performed using Scikit-learn version 0.23.1. The optimal number of clusters in the datasets was determined by ranging $k$ from 2 to 6 and selecting the number of clusters with the maximum silhouette score[101]. Based on the clusters formed in the previous step DEG analysis was performed using the limma-voom pipeline[104] of the R edgeR package version 3.32.1. Bi-clustering was performed on the DEGs using the iterative search algorithm (ISA). ISA requires initial seeding vectors both on genes and samples. These vectors were generated by first doing hierarchical clustering using the hclust function in R with the method set to ward.D resulting in 10 gene clusters and 7 sample clusters. From these clusters, 3 samples in each cluster were selected randomly and used to create the seeding vectors. Because ISA is stochastic in nature, the results can vary based on seeding vectors, thus we ran the algorithm 25 times and only bi-clusters that appeared in more than 75% of the trials were considered legitimate (Supplementary Fig. 12). Kaplan–Meier analysis was then done between each bi-cluster and the remainder of the samples. To assess whether bi-cluster gene signatures based on the canine data were applicable to the human TARGET dataset, K-Means clustering was performed with optimal K, ranging from 2 to 6, chosen by selecting the maximum silhouette score and then followed by Kaplan–Meier analysis.

**Immunohistochemical (IHC) analysis**. All canine tissues were fixed with 10% neutral buffered formalin, decalcified using 12% EDTA with a pH of 7.2, and subsequently embedded in paraffin. An antibody panel against specific immune cell markers (Supplementary Tables 1, 2) was used to stain a subset of 20 specimens. Labeling of canine tissues for CD204 was completed by the Animal Health Diagnostic Center at Cornell University. The remaining canine tissue IHC was completed by the Histology Laboratory at the University of Georgia, College of Veterinary Medicine. Immunohistochemical labeling was examined at 400x (hpf; 0.196 mm$^2$) using an Olympus CX43 microscope. Cells expressing Iba1 and CD204 were numerous and scored semi-quantitatively as 1+ (0–50/hpf), 2+ (51–100/hpf), or 3+ ( >100/hpf). Cells expressing CD3, CD45RA, FOXP3, CD20, and MUM1 were similarly scored as 1+ (<1/hpf), 2+ (1-5/hpf), or 3+ (>5/hpf). Toluidine blue staining was completed by VitroVivo (Supplementary Fig. 10) using the VitroView Toluidine Blue Stain Kit (VB-3013). Two canine tumors were excluded from the immunohistochemical analysis due to poor tissue quality and high background labeling. Five human OS samples, decalcified with formic acid, were labeled and examined for expression of CD3, CD20, and CD204 (Supplementary Table 2).

**Statistics and reproducibility**. For Kaplan–Meier analysis, the log-rank test was computed by the R function ggsurvplot survminer version 0.4.90. P-values or false discovery rates (FDR) less than 0.05 were considered significant. All statistics were corrected for multiple testing using Benjamini Hochberg (BH) method either provided as an option by the software package used or using the p.adjust function in R. Details on the calculation of GSEA FDR can be found in Subramanian et al.[102]. A two-tailed Mann–Whitney U-test was used to evaluate CIBERSORTx scores among clusters. Overrepresentation Analysis (GO analysis) was performed on the gene sets with the PANTHER platform using a Binomial test and FDR correction[110,111] The Mann–Whitney U-test was used to perform statistical analysis on the IHC staining with a p-value of less than 0.05 being considered significant.

**Reporting summary**. Further information on research design is available in the Nature Portfolio Reporting Summary linked to this article.

# Data availability

Human osteosarcoma data obtained from the TARGET Childhood Cancer Program is openly available from NCI Genomic Data Commons Portal. Canine mRNAseq data were deposited into the Gene Expression Omnibus database under accession number GSE238110 and are available at the following URL: https://www.ncbi.nlm.nih.gov/geo/query/acc.cgi?acc=GSE238110.

# Code availability

Code can be found at https://doi.org/10.5281/zenodo.8125077 and is also provided in the Supplementary Information as Supplementary Note 1. Standard algorithms were used as reported in Methods. All software packages used are open source, except for Ingenuity Pathway Analysis, and are freely available to the public.

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

## Acknowledgements

RNA sequencing and initial data analysis were conducted at the Frederick National Laboratory for Cancer Research (FNLCR) at the CCR Sequencing Facility, NCI, NIH, Frederick, MD 21701. This work was supported by the Intramural Program of the National Cancer Institute, NIH (Z01-BC006161) and the Intramural Research Programs of the National Center for Advancing Translational Sciences, NIH (Z01-TR000249). The content of this publication does not necessarily reflect the views or policies of the Department of Health and Human Services, nor does mention of trade names, commercial products, or organizations imply endorsement by the U.S. Government. The funders had no role in study design, data collection, and analysis, decision to publish, or preparation of the manuscript. The authors thank Joseph Meyer for assistance with graphical illustrations.

## Author contributions

J.D.M., G.T., D.G., J.B., A.K.L., J.A.B., P.M., and T.A.M. designed and supervised the experiments and data analyses. J.D.M., C.M., J.A.B., C.M.S., T.A.M. performed

experiments and/or analyzed the data. T.A.M., D.G., P.M., J.B., and G.T. provided insight into the presentation of data in display items. A.K.L., J.D.M., and J.A.B. wrote the manuscript. All authors discussed the results, revised the draft manuscript, and read and approved the final manuscript.

## Funding

## Competing interests
The authors declare no competing interests.

## Additional information

 ns license, unless indicated otherwise in a credit line to the material. If material is not included in the article's Creative Commons license and your intended use is not permitted by statutory regulation or exceeds the permitted use, you will need to obtain permission directly from the copyright holder. To view a copy of this license, visit http://creativecommons.org/licenses/by/4.0/.

This is a U.S. Government work and not under copyright protection in the US; foreign copyright protection may apply 2023

