## [Peer Review File · Communications Biology]

Reviewers' comments:

Reviewer #1 (Remarks to the Author):

RNA seq data was compared to osteoblasts. It is unclear if this is the best comparator. It would also be appropriate to identify the cell line in the methods as CnOb. In addition, no information regarding passage number or validation of the cell line were included in either this manuscript or the Gardner reference cited for its use.

The further differentiation of the previously published gene signatures into potentially differing outcomes within each GS is both significant and interesting.

The identification of the same pattern gene signature patterns within the human TARGET database further supports the applicability of the canine model.

CIBERSORTx data should be viewed with caution as the deconvolution of cell types requires assumptions about gene expression that may not be applicable to those cells in novel situations and/or environments. The relative quantitation of some immune cell markers does reassure that the differences noted are likely real. More precise quantitation of immune cells populations would further confirm that the DGE analysis predicts actual cell populations. Some of the issues with CIBERSORTx are also addressed by the authors in the discussion.

Reviewer #2 (Remarks to the Author):

The authors generated a canine OS RNA-Seq dataset from a clinical trial where clinical data is available. The authors use two prior gene sets to separate the new dataset into groups and claim that these groups are associated with differential outcomes. This analyses was repeated with a human dataset.

It is unclear how groups were assigned into poor outcome and good outcome groups following clustering in figure 1 and figure 2. This is essential to evaluate whether the outcome associations are meaningful. This especially important/concerning because the clustering dendrograms do not appear too to be the main driver of the groupings shown in figure 1 and figure 2.

The authors then use GSEA to look for differentially expressed genes between good and poor outcome groups and identify that these genes are enriched with immune genes. (Figure 3)

It is my opinion that directly using the outcome to define these groups (rather than the GS1 and GS2 defined groups) would be a better way to understand these enrichment patterns.

The authors then use cybersort deconvolution to look at differential immune cell subtypes between good and poor outcome groups as defined by GS1 and GS2.

The authors then stain a number of tumors to show that variable immune infiltrate is present in human and canine tumors.

Overall it is my opinion that the paper would be much stronger if the patterns of expression associated with outcomes were derived from the canine dataset itself rather than inferred from previously described gene sets based on the differences between osteoblasts and tumors.

I would be quite surprised if the comparison of osteoblasts(no immune cells) to tumors(contain immune cells) did not define the absence of immune cells in the osteoblast samples.

Many previous papers have shown that immune infiltration is important to OS outcome using a variety of datasets. Many papers have shown that transcriptional signatures correspond to pathology for immune infiltration.

This is likely a very useful dataset, but the analyses done here does not generate a cohesive novel story and I don't think that I could repeat it- specifically the group designations bases on what is presented.

I wish I could be more favorable as I agree strongly with the authors that comparative oncology using canine tumors is an important approach to study osteosarcoma.

Reviewer #3 (Remarks to the Author):

Osteosarcoma (OS) is one of the most common bone tumor in the pediatric population. Canine OS are highly comparable with their human counterpart, so it is a suitable pre-clinical model for understanding the pathophysiology as well as the future therapeutic approaches for the disease. In this study, Mannheimer et al. their goal is to identify distinct molecularly defined patient subsets and/or prognostic gene signatures among dogs receiving standardized therapy, and to evaluate if and how these signatures are translatable to human OS using publicly available data. The authors confirmed through the computational model and with the patient samples that distinct immune cell populations play a significant role in the outcomes of canine OS. The author also established similar transcriptional and clinical patterns for human disease and argued for using canine-omics data to discover new therapeutic targets and treatment strategies for human OS. The results would be helpful for future anti-tumor therapy, particularly for osteosarcoma. The manuscript is of interest to the osteosarcoma research community. There are some concerns noted below:

General Comment: It would be easy for the reader if you provided a flow-chart demonstrating the sequential use of tools in evaluation.

1. Abstract is a little vague...would prefer more specifics presented here about their findings...what about T cells and Mos.
2. The authors mention: that identifying subsets of dogs that might serve as candidates for therapeutic testing? What does this mean? Clarification would be helpful.
3. How is this dataset different from Gardner et al,---which also showed comparable populations. This reviewer was wondering if it was essential to use the signatures previously identified from Gardner et al.? Could the authors determine own signatures from their data? Could be productive, and more novel to determine own transcriptomic signatures.
4. Significant differences between human and canine. What are the global overlapping signatures between TARGET human and canine?
5. Not exactly clear about the analysis of the of the GS-1 and GS2 with the localized TAREGT pts. What trying to demonstrate here?
6. Fig 1F: Really statistically significant...lines cross/overlap by 2 years. Discuss/Comment.
7. Overall these GS signatures not really demonstrate a large difference in survival that would lead to drastic identification of molecular events that can make a difference therapeutically?
8. Besides these GS's, can you find a more differential group of canines that survive and deceased at 1 and/or 2 years...then profile those DEGs? Again, referring to identifying your own signature from

this large dataset with well annotated clinical correlations.

9. Please comment on human and canine tumors' distinct activated mast cell populations. Also, will it affect the patient therapeutic outcomes? Further elaboration/discussion on these differences between human and canine is necessary.

10. The author observed macrophages as one of the most abundant immune populations in the TME of osteosarcoma. The author also observed that M2-macrophage populations were associated with a good prognosis that might be unique to the bone tumor. The author hypothesized that since both macrophage and osteoclast are derived from the monocytes, it is possible that a shift in monocyte differentiation to M2 macrophages reduces osteoclast formation and thereby inhibits bone resorption, favor the role of osteoclast in tumor development. Please provide evidence by analyzing the osteoclast (OCs) population in the TME of osteosarcoma in the patient samples. Does the OCs abundance associate with poor outcomes?

11. Typo error 'Comparison of the immunohistochemically analysis to the GS-1 signature can be seen in Figure 4-A instead of Figure 6-A.

12. Fig. 6: quantify the IHC...1+ to 3+? Stats?

13. For Supp Fig 3...what are the gene signatures and Pathways associated with the differential survival from GS1TARGET

NATIONAL CANCER INSTITUTE
Center for Cancer Research

Amy K. LeBlanc DVM | Director, Comparative Oncology Program
10 Center Dr. Room 1B53
Bethesda, MD 20892

May 17, 2023

Dear Editorial Staff,

Please find enclosed an electronic version of our revised original Research Article COMMSBIO-22-4011 entitled, "Transcriptional profiling of canine osteosarcoma identifies prognostic gene expression signatures with translational value for humans".

In line with the thorough and insightful comments provided by the reviewers, the manuscript and associated display items have been extensively revised. Of note, an experimental schema has been added (new Figure 1) and *de novo* gene signatures were devised from the canine transcriptional data and applied to both canine and human datasets. Specific point-by-point responses to all reviewer comments are provided in a separate file. The CIBERSORTx data is now presented as a Table and highlighted in Figures 6 and 7.

We thank the Editorial staff and reviewers again for their careful critique of the work and look forward to additional suggestions on how to improve this submission for consideration of publication.

Sincerely yours,

Amy K. LeBlanc DVM DACVIM (Oncology)
Director, Comparative Oncology Program
Center for Cancer Research, National Cancer Institute
Bethesda, MD 20892
amy.leblanc@nih.gov
240-760-7093

REVIEWERS' COMMENTS:

Reviewer #1 (Remarks to the Author):

The authors have responded to the concerns of the reviewers and I have no further comments.

Reviewer #3 (Remarks to the Author):

The authors have satisfactorily addressed the comments and critiques.